# Pyrenebutyrate Enhances the Antibacterial Effect of Peptide-Coupled Antisense Peptide Nucleic Acids in *Streptococcus pyogenes*

**DOI:** 10.3390/microorganisms11092131

**Published:** 2023-08-22

**Authors:** Corina Abt, Lisa Marie Gerlach, Jana Bull, Anette Jacob, Bernd Kreikemeyer, Nadja Patenge

**Affiliations:** 1Institute of Medical Microbiology, Virology and Hygiene, University Medicine Rostock, 18057 Rostock, Germanyjana.bull@med.uni-rostock.de (J.B.); bernd.kreikemeyer@med.uni-rostock.de (B.K.); 2Peps4LS GmbH, 69120 Heidelberg, Germany; jacob@peps4ls.de

**Keywords:** pyrenebutyrate, antimicrobial activity, antimicrobial therapy, antisense molecules, *Streptococcus pyogenes*, *Streptococcus pneumoniae*, *Escherichia coli*, *Klebsiella pneumoniae*, peptide nucleic acid, PNA, cell-penetrating peptide, CPP

## Abstract

Antisense peptide nucleic acids (PNAs) inhibit bacterial growth in several infection models. Since PNAs are not spontaneously taken up by bacteria, they are often conjugated to carriers such as cell-penetrating peptides (CPPs) in order to improve translocation. Hydrophobic counterions such as pyrenebutyrate (PyB) have been shown to facilitate translocation of peptides over natural and artificial membranes. In this study, the capability of PyB to support translocation of CPP-coupled antisense PNAs into bacteria was investigated in *Streptococcus pyogenes* and *Streptococcus pneumoniae*. PyB enhanced the antimicrobial activity of CPP-conjugated antisense PNAs in *S. pyogenes*. The most significant effect of PyB was observed in combination with K8-conjugated anti-*gyrA* PNAs. In contrast, no significant effect of PyB on the antimicrobial activity of CPP-conjugated PNAs in *S. pneumoniae* was detected. Uptake of K8-FITC into *S. pyogenes*, *Escherichia coli*, and *Klebsiella pneumoniae* could be improved by pre-incubation with PyB, indicating that PyB supports the antimicrobial effect of CPP-antisense PNAs in *S. pyogenes* by facilitating the translocation of peptides across the bacterial membrane.

## 1. Introduction

Peptide nucleic acids (PNAs) are synthetic DNA analogues with a pseudo-peptide backbone consisting of N-(2-aminoethyl)glycine units that are linked by amide bonds. Nucleobases are bound to the polyamide backbone via methylene carbonyl linkages. PNAs are able to hybridize with high affinity to complementary DNA and RNA molecules in a sequence-specific and reversible manner. PNA binding is independent of the salt concentration and the molecules are chemically stable [1]. Since PNAs are not recognized by proteases and nucleases, they exhibit a high serum stability [2]. Strong specific binding in combination with a high stability make PNAs a suitable tool for gene silencing. Antimicrobial activity is achieved by targeting essential bacterial genes. Typical genes are involved in central metabolic pathways that are in many cases also classical targets of conventional antibiotics. Examples include gene coding for subunits of the RNA polymerase (*rpoD*) or gyrase (*gyrA*), cell wall synthesis enzymes (*murA*), or enzymes involved in the synthesis of fatty acids (*acpP*, *fabI*) [3,4,5,6,7,8,9]. However, surface barriers, including membranes, cell walls, and capsules, are a hindrance to translocation of PNAs into bacteria. Uptake of PNAs into bacteria can be improved by coupling with cell penetrating peptides (CPPs). These molecules have been intensively investigated in the context of cargo delivery into eukaryotic cells [10].

The uptake of CPP–PNA into bacterial cells depends on surface characteristics and therefore differs between species, strains, and even serotypes. CPP-antisense PNAs exhibit antibacterial activity towards bacteria. Over recent years, antisense inhibition of Gram-negative bacteria has been investigated in detail [4]. In contrast, the number of CPPs supporting PNA uptake into Gram-positive bacteria is low. CPPs facilitating bacterial PNA uptake have been identified for *Listeria monocytogenes* and *Staphylococcus aureus* [3,11]. Recently, we identified CPPs efficiently promoting the uptake of antisense PNAs in *Streptococcus pyogenes* and *Streptococcus pneumoniae*. We identified three different CPPs that enhance the antibacterial effect of antisense PNAs in *S. pyogenes*-targeting *gyrA:* HIV-1 TAT, K8, and (RXR)_4_XB [12]. HIV-1 TAT and (RXR)_4_XB were suitable CPPs for antisense PNA delivery in *S. pneumoniae* [13].

An improved uptake of CPP-antisense PNA conjugates will increase their antibacterial activity and allow the application of lower CPP–PNA concentrations. This may lead, for example, to a considerable cost reduction in gene expression silencing experiments. Addition of counter anions has been shown to facilitate translocation of arginine-rich CPPs across intact lipid bilayer membranes [14,15]. One example that has already been studied for this approach is pyrenebutyrate (PyB), which is obtained from 4-(1-pyrenyl)-butyric acid [16,17]. PyB is hydrophobic due to its aromatic ring system and has a negatively charged carboxylic group. An improved uptake of oligoarginine–cargo conjugates into various cell lines has been demonstrated using cell fluorometry and confocal microscopy and confirmed by bioactive cargo delivery experiments [16,17]. Competition with various ionic species in complex solutions such as media and sera impedes the complex formation of arginine-rich peptides with PyB [17]. Therefore, the application of this counterion is limited to in vitro experiments in the absence of complex solutions.

Overall, PyB has been shown to increase the translocation efficacy of arginine-rich CPP-cargo conjugates in various cell lines, but translocation of CPP-coupled cargos has not yet been investigated in bacteria. In contrast to mammalian cells, bacteria produce polymeric surface structures that provide barriers to both CPPs and PyB, e.g., peptidoglycan-based cell walls or capsules composed of polysaccharides or proteins. Additionally, the plasma membrane of bacteria differs from the mammalian cell membrane in its transmembrane potential and lipid composition. Bacterial membranes contain a high amount of negatively charged phospholipids such as cardiolipin and phosphatidylglycerol [18], while mammalian membranes are composed of mostly neutral lipids such as cholesterol, phosphatidylcholine and sphingomyelin [19].

These structural differences between mammalian membranes and bacterial surfaces raise the question whether PyB is able to support translocation of peptides into bacterial cells. Here, we investigated the effect of PyB on CPP-coupled antisense PNA activity in the Gram-positive bacteria *S. pyogenes* and *S. pneumoniae*. PyB administered alone showed no toxic effect on the bacteria. Pre-incubation of *S. pyogenes* M49 strain 591 with PyB enhanced the antimicrobial activity of CPP-coupled antisense PNAs. In contrast, the effect of CPP-antisense PNAs in *S. pneumoniae* was not significantly influenced by PyB. Uptake of K8-FITC by *S. pyogenes* and the Gram-negative bacteria *Escherichia coli* and *Klebsiella pneumoniae* increased in the presence of PyB. These results indicate that PyB enhances the antimicrobial effect of CPP-antisense PNAs in *S. pyogenes* by supporting CPP translocation over the bacterial membrane.

## 2. Materials and Methods

### 2.1. PNA Synthesis

CPP–PNAs were synthesized on RinkAM resin using Fmoc chemistry followed by HPLC purification (Peps4LS GmbH, Heidelberg, Germany) [20]. The ethylene glycol linker (Fmoc-8-amino-3,6-dioxaoctanoic acid) and the respective Fmoc-protected amino acids for the CPPs were coupled directly to the PNA part on the resin. Sequences of all CPP–PNAs used in this study are listed in Table 1.

### 2.2. Bacterial Strains and Culture Conditions

*S. pyogenes* serotype M49 strain 591 [21] was cultured in Todd–Hewitt broth supplemented with 0.5% yeast extract (THY; Oxoid, Wesel, Germany) at 37 °C under a 5% CO_2_ atmosphere. *S. pneumoniae* serotype 4 strain TIGR4 [22] was grown on Columbia blood agar plates (Becton Dickinson, Heidelberg, Germany) and cultivated for 5 h in Brain Heart Infusion broth (BHI, Oxoid, Wesel, Germany) at 37 °C under a 5% CO_2_ atmosphere. *E. coli* ATCC 35218 and *K. pneumoniae* ATCC 13883 were cultivated in Luria–Bertani (LB) broth at 37 °C and 200 rpm. Viability was verified by CFU/mL determination on agar plates.

### 2.3. Bacterial Killing Assay

*S. pneumoniae* serotype 4 strain TIGR4: after culturing, *S. pneumoniae* serotype 4 strain TIGR4 was diluted to approximately 10^5^ CFU/mL in phosphate-buffered saline (PBS)/THY (95%/5%). *S. pyogenes* serotype M49 strain 591: An overnight culture of *S. pyogenes* serotype M49 strain 591 was diluted to approximately 10^5^ CFU/mL in PBS/BHI (70%/30%). An amount of 400 µL of the respective bacterial suspension was transferred to a 2 mL polypropylene reaction tube. 4-(1-pyrenyl)-butyric acid (Merck, Darmstadt, Germany) was dissolved to 1 M in DMSO. The solution was stored in the dark and further dilution steps were performed in PBS and 3% DMSO. An amount of 50 µL PyB in PBS/DMSO was added to the bacterial suspension at a final concentration of 50–400 µM and pre-incubated for 10 min at 37 °C and 7 rpm (Rotor SB3, Stuart, Staffordshire, UK). An amount of 50 µL PBS served as a mock control. Following pre-incubation, 50 µL of CPP–PNA solution was added to a final concentration of 5–10 µM or as indicated. An amount of 50 µL molecular grade H_2_O served as a mock control. The reaction tubes were incubated for another 6 h at 37 °C and 7 rpm. Killing kinetics were observed over 12 h. Samples were collected at time points of 2–12 h following addition of CPP–PNAs, and bacteria were plated on BHI solid medium following serial dilution. Agar plates were incubated for 24 h at 37 °C under a 5% CO_2_ atmosphere and CFUs were counted. The bacterial count at the beginning of the experiment was 1–3 × 10^5^ CFU/mL. Each experiment was conducted at least three times in independent biological replicates.

### 2.4. Fluorescence Peptide Uptake Assay

*S. pyogenes*, *S. pneumoniae*, *E. coli*, or *K. pneumoniae*, respectively, were pre-incubated with 100 µM PyB. Bacteria were then incubated with 10 µM K8-FITC (Eurogentec, Seraing, BE) for 30 min at 37 °C, washed in PBS, and lysed. For lysis, bacteria were incubated in lysis buffer (Quiagen, Hilden, Germany). *S. pyogenes* samples were supplemented with Mutanolysin (Merck, Darmstadt, Germany). Cell debris was removed by centrifugation. Afterwards, the fluorescence intensity of cell lysates was measured with a fluorescence microplate reader in arbitrary units at excitation/emission wavelengths of 488 nm/521 nm. Cell lysates of bacteria treated with H_2_O with or without pre-treatment with 100 µM PyB served as background control.

### 2.5. Statistical Analyses

All experiments were performed at least three times or as indicated in the respective figure legends by sample size (n). Statistical significance was determined using the tests indicated in the respective figure legends. Statistical analyses were performed using GraphPad Prism (version 7, GraphPad Software, Boston, MA, USA).

## 3. Results

### 3.1. Effects of Pyrenebutyrate on the Activity of CPP-Coupled Antisense PNAs in S. pyogenes and S. pneumoniae

Recently, we identified peptide-coupled antisense PNAs specific for *S. pyogenes* and *S. pneumoniae* that exhibited antibacterial activity [12,13]. Since hydrophobic counterions such as pyrenebutyrate (PyB) have been shown to support translocation of peptides over membranes, we wanted to investigate whether PyB is able to support uptake of antisense CPP–PNAs into *S. pyogenes* and *S. pneumoniae*. PyB has only been studied in eukaryotes but not in bacteria. Therefore, the toxicity of PyB towards bacteria was determined in a bacterial killing assay [23].

A total of 5 × 10^5^ CFU/mL *S. pyogenes* 591 or 8 × 10^5^ CFU/mL *S. pneumoniae* TIGR4 were incubated with PyB concentrations from 50 µM to 200 µM or 50 µM to 400 µM, respectively, for 6 h and the bacterial count was determined. The concentrations tested are based on the doses of PyB applied in similar experiments in cell cultures [17]. CFU/mL values following incubation with PyB were compared to an untreated control (Figure 1). No significant toxicity of PyB was observed towards *S. pyogenes* up to a concentration of 200 µM PyB (Figure 1A). *S. pneumoniae* was unaffected by PyB up to a concentration of 400 µM (Figure 1B). Since viable counts decreased slightly after treatment of *S. pyogenes* with 200 µM PyB, 100 µM PyB was used for treatment of *S. pyogenes* and 200 µM PyB was used for treatment of *S. pneumoniae* in the following experiments. To study the effects of PyB on the antimicrobial activity of CPP–PNA, bacteria were pre-incubated for 10 min with PyB.

The antisense-PNA target used in this study was the essential gene *gyrA*, which encodes the subunit A of the DNA gyrase. Peptides were coupled to PNAs via a flexible ethyleneglycol linker (8-amino-3, 6-dioxaoctanoic acid). The sequences of all peptide-conjugated PNAs used in this study are listed in Table 1. Antisense PNAs were designed to be complementary to the start region of the target gene. Scrambled control PNAs (scrambled PNAs, scPNAs) are composed of the same base pairs as the antisense PNAs but cannot bind to the target mRNA because of their randomized order. All constructs have been used previously to study the antimicrobial effects of antisense PNAs in the respective streptococci [12,13].

Since previous studies with eukaryotic cells only determined the effects of PyB with arginine-rich CPPs, the numbers of positive amino acid residues of the carriers studied in this work are listed in Table 2 for comparison.

*S. pyogenes* M49 strain 591 was incubated for 6 h with CPP-antisense PNA conjugates with or without pre-treatment with 100 µM PyB for 10 min. A reduction in the bacterial CFU/mL caused by 5 µM of different CPP-coupled anti-*gyrA*-PNA constructs compared to the untreated control was observed. Antisense PNAs coupled to three different CPPs were tested: (RXR)_4_XB, HIV-1 TAT (TAT), and Oligolysin (K8) (Table 1). The number of CFUs in CPP–PNA-treated samples was significantly reduced in comparison to the untreated control sample (Figure 2A). Pre-treatment with PyB resulted in a greater reduction in bacterial counts. The effect increased as a function of the positive charge of the CPPs. Treatment with K8-anti-*gyrA*-PNA caused a log CFU reduction of 3.7 that was significantly enhanced to 4.8 following pre-treatment with PyB.

Incubation of *S. pneumoniae* TIGR4 for 6 h with 10 µM (RXR)_4_XB- and TAT-conjugated anti-*gyrA*-PNA (Table 1) was performed with or without pre-treatment with 200 µM PyB for 10 min (Figure 2B). As observed before [13], the CPP (RXR)_4_XB supports uptake of antisense PNAs more efficiently than TAT in *S. pneumoniae*. Pre-treatment with PyB did not significantly enhance the CFU reduction in this experiment. K8-coupled antisense PNA did not affect *S. pneumoniae* in previous experiments [13]. Pre-treatment with PyB was not able to increase the efficacy of this CPP. We also tested Oligoarginine- (R9) and PenArg-coupled anti-*gyrA*-PNAs, which caused a log CFU reduction of 4 and 3.3, respectively. There was also no significant effect of PyB treatment on the antimicrobial efficacy of these constructs. A slight, albeit non-significant, improvement in the antimicrobial effect of CPP–PNA on *S. pneumoniae* mediated by PyB depends on the proportion of positively charged amino acid residues of the carriers. In pneumococci, the activity of CPP–PNAs containing less than 75% arginine and lysine ((RXR)_4_XB and PenArg, Table 2) was not enhanced by PyB, whereas the conjugates TAT-anti-*gyrA* PNA (75%) and R9-anti-*gyrA* PNA (100%) (Table 2) showed slightly increased antibacterial activities in the presence of PyB. Overall, PyB did not significantly enhance the antimicrobial activity of CPP–PNA constructs in *S. pneumoniae* under the conditions used in this study.

### 3.2. Bactericidal Kinetics of CPP-Coupled Antisense PNAs in the Presence of PyB in S. pyogenes

In *S. pyogenes*, addition of PyB resulted in the significantly increased antimicrobial activity of K8-anti-*gyrA* PNA. Therefore, the kinetics of the antimicrobial activity of K8-conjugated anti-*gyrA* PNA in the presence of PyB were studied in an in vitro time killing assay in *S. pyogenes*. Bacteria were treated with 2.5 or 5 μM CPP-antisense PNAs or scrambled control PNAs (Figure 3) with or without pre-treatment with 100 µM PyB. Samples were collected at 2, 4, 6, and 12 h following antisense treatment. CFUs per milliliter were determined by plating of serial dilutions and plotted over time. Untreated bacteria grew by 1–1.5 log CFU over the course of the experiment. Incubation with K8-anti-*gyrA* PNA resulted in a constant decrease in viable bacteria over time. However, *S. pyogenes* was not eradicated during the course of the assay (Figure 3A,B). In contrast, following pre-treatment with PyB, application of 5 µM K8- anti-*gyrA* PNA *S. pyogenes* CFUs steadily declined until clearance after 12 h (Figure 3B). Following treatment with 2.5 µM K8-conjugated scPNA, no reduction in bacterial counts was observed throughout the experiment. Upon treatment with 5 µM K8-conjugated scPNA, growth was reduced by 0.6 log CFU in comparison to the untreated control, indicating an inhibitory effect of the CPP under these conditions. Overall, pre-treatment with PyB enhanced the activity of K8-anti-*gyrA* PNA and allowed the eradication of *S. pyogenes* in vitro.

### 3.3. PyB Supports Transport of K8-FITC Peptides into Bacteria

Since PyB enhanced the antibacterial effect of K8-anti-*gyrA* PNA in *S. pyogenes*, we hypothesized that uptake of K8 was facilitated in the presence of PyB. Therefore, we investigated uptake of K8-FITC into *S. pyogenes* with or without pre-treatment with 100 µM PyB for 10 min. Additionally, to study whether the uptake of K8-FITC is also facilitated by PyB in Gram-negative bacteria, we pre-incubated *E. coli* and *K. pneumoniae* with 100 µM PyB for 10 min before incubation with K8-FITC. Bacteria were incubated with 10 µM K8-FITC for 30 min, washed in PBS, and lysed. Subsequently, the fluorescence intensity of the bacterial cell lysates was measured with a fluorescence microplate reader in arbitrary units at excitation/emission wavelengths of 488 nm/521 nm. The fluorescence intensity of cell lysates from *S. pyogenes* incubated with K8-FITC increased 4.5-fold when the cells were pre-treated with PyB (Figure 4).

It can therefore be concluded that pre-incubation of *S. pyogenes* with PyB increases the uptake of PNA conjugates depending on the carrier. The same effect was observed in *E. coli* and *K. pneumoniae*, with a fluorescence intensity that increased 6-fold and 5-fold, respectively (Figure 4). These data support the idea that counterions enhance the bacterial uptake of CPP-coupled cargos.

## 4. Discussion

Antisense molecules as potential therapeutic agents provide species specificity due to the free choice of the target gene and the sequence variability of a given gene between species. To date, no naturally occurring resistance of bacteria to antisense oligonucleotides has been observed. One advantage of antisense molecules is the possibility of easily modifying the antisense sequence of the molecule once the pathogen in question develops resistance. Antisense-PNAs are potentially interesting antisense compounds, which are currently under intensive investigation. One challenge to their successful application is the limited translocation of PNAs into bacteria. CPPs are coupled to many cargos, including PNAs, to enhance their uptake into bacterial and eukaryotic cells [24,25].

The counterion PyB can be employed as a translocation catalyst, accelerating the direct translocation of arginine-rich CPPs in various cell lines [16,17]. In artificial model membranes, the effects of PyB on R8 translocation were studied. PyB significantly accelerated the accumulation of R8 on membranes containing negatively charged lipids, leading to the internalization of R8. PyB also increased the fluidity of the negatively charged membranes and induced membrane curvature [26]. In this work, the effect of PyB on the antimicrobial activity of CPP–PNAs in bacteria was investigated. In the presence of PyB, uptake of K8-FITC conjugates into *S. pyogenes*, *E. coli*, and *K. pneumoniae* was strongly elevated, and the antibacterial effect of CPP–PNA on *S. pneumoniae* and *S. pyogenes* was increased. Thus, this work demonstrated that the enhancing impact of PyB on translocation is not limited to eukaryotes. PyB is also effective in bacteria, despite differences in membrane composition and charge and the presence of bacterial barriers such as the peptidoglycan layer, LPS, and capsules. The effective concentration of PyB in the tested streptococcal species was comparable to those used in cells [16,17]. PyB alone had neither intrinsic effects on bacterial viability at the concentrations used nor did it increase the toxic effects of CPP–scPNA controls.

The strongest effect of PyB on the antimicrobial activity of CPP–PNAs was observed with K8-*anti gyrA* PNA in *S. pyogenes*. Until now, it was assumed that the guanidinium cations of arginine residues of CPPs are particularly important for improved uptake via counter anions [26,27,28]. Oligolysine K8 is a cationic CPP without guanidinium moieties. Therefore, interactions of PyB with guanidinium cations of the CPPs do not appear to be essential to increase the translocation of CPPs into bacteria. It seems likely that the interaction of positively charged CPPs with negatively charged PyB is sufficient to increase the accumulation of CPPs on the membrane surface, thereby increasing the efficiency of CPP uptake. Furthermore, it has been shown that PyB enhances the direct translocation of oligoarginines through artificial membranes not only directly due to electrostatic interaction with the CPP, but also indirectly by increasing membrane fluidity and the formation of membrane curvatures [26]. Eukaryotic cell and bacterial membranes are not homogenous in composition and charge, but some areas are more fluid than others. As a rule, the translocation rate of cationic CPPs is higher in fluid membrane domains [29,30,31]. Incubation with PyB was shown to disperse phase separation and increase the fluidity of negatively charged artificial membranes, which might result in an enhanced translocation of cationic CPPs. Additionally, PyB caused formation of inward tubular structures in artificial membranes, which disappeared upon addition of oligoarginine. Formation of curvatures and recovery might enhance the uptake of CPPs [26]. Whether one of these effects of PyB demonstrated with artificial membranes or the combination of these effects enhances uptake of positively charged oligopeptides in bacteria is not known. It has also not yet been tested whether PyB increases the uptake of oligolysine not only in *S. pyogenes* but also in eukaryotic cells.

The effect of PyB was stronger in *S. pyogenes* than in pneumococci. The efficiency of different carriers varies generally between the two streptococcal species, for example, K8 is a very efficient CPP in *S. pyogenes* but does not support translocation in pneumococci [13]. Surface structures such as the capsule play a role in the uptake of carriers [6,32,33,34] and might also influence the activity of the counterion. Pneumococci generally have a thick capsule, and particularly the capsule of *S. pneumoniae* TIGR4 is quite wide at 185 nm [35,36]. In addition to thickness, the capsules of individual streptococcal species also differ in composition. *S. pyogenes* capsules are composed of hyaluronic acid (N-acetlyglucosamine and glucuronic acid), whereas pneumococcal capsules consist of 2–8 saccharides that are often substituted with O-acetyl, phosphoglycerol, or pyruvyl acetal residues [37,38,39,40].

PyB application is limited by the fact that CPP delivery is not efficiently supported in the presence of culture medium or serum. It has been speculated that ionic species in the medium compete with the carrier for interaction with PyB [17]. Use of a counterion is consequently restricted to in vitro approaches, in which the experimental conditions can be adapted in a way that the CPP–PNA uptake can be increased. If the required concentration of CPP–PNA can be decreased, the expenses for in vitro studies of the antisense PNA function will be reduced. To enable the use of PyB in vivo, Mishra and coworkers tested an approach in which they covalently coupled PyB to a CPP–cargo molecule. Compared to CPP alone, the uptake of the PyB–CPP-cargo conjugates into cells was increased 4–5-fold and was shown to be independent of the medium [41]. In future experiments, it could be tested whether linking PyB to CPP–PNA might result in a conjugate with an improved activity in bacteria.

The fluorescence intensity of cell lysates from *S. pyogenes*, *E. coli*, and *K. pneumoniae* incubated with K8-FITC increased 4-fold, 6-fold, and 5-fold, respectively, when the cells were pre-treated with PyB. A similar effect of PyB has been observed on the uptake of GABA-FITC-labelled R9 peptide in HeLa cells [16]. In the same study, the impact of PyB on the relative uptake of labelled pTat 48–60 and Pen was much lower. In conclusion, the efficacy of the translocation-enhancing effect of PyB depends on the properties of the target cell as well as on the nature of the CPP used. Based on our results, it should be investigated whether PyB is able to enhance the uptake of CPP–PNAs in Gram-negative bacteria.

## 5. Conclusions

Overall, we demonstrated that PyB supports the antimicrobial effect of CPP-coupled antisense PNAs in *S. pyogenes*. The effect of PyB was not limited to arginine-rich CPPs, but PyB was able to enhance the activity of PNAs coupled to lysine-rich CPPs. Pretreatment of *S. pyogenes*, *E. coli*, and *K. pneumoniae* with PyB improved the cellular uptake of FITC-labelled K8. Together with data from previous studies in eukaryotic cells and artificial membranes [14,16,17], these results indicate that PyB directly interacts with the bacterial membrane and thereby facilitates CPP uptake. In future studies, more potential enhancing counterions should be identified, which may promote membrane translocation in the presence of complex media and sera. Alternatively, covalent coupling of PyB to CPPs could be investigated as a measure to optimize PyB-supported uptake.

## Figures and Tables

**Figure 1 microorganisms-11-02131-f001:**
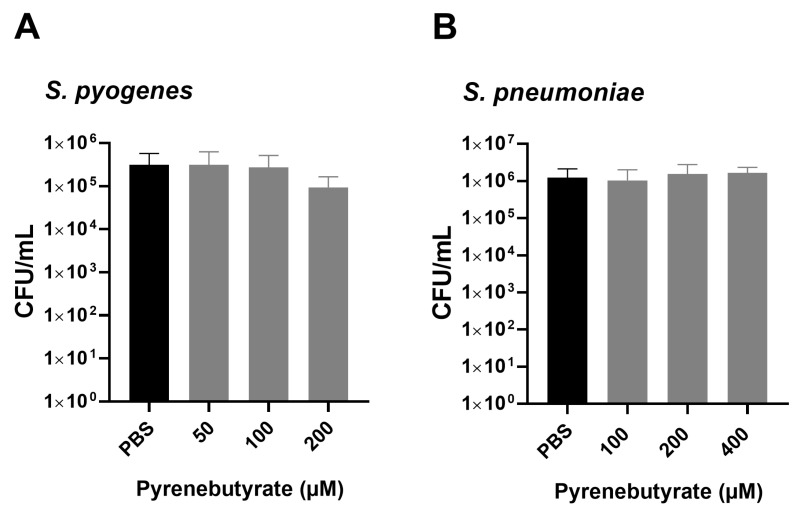
Toxicity of PyB to *S. pyogenes* and *S. pneumoniae*. (**A**) Treatment of *S. pyogenes* with 50–200 µM PyB for 6 h. (**B**) Treatment of *S. pneumoniae* with 50–400 µM PyB for 6 h. Data are presented as mean values with standard deviation. Statistical significance was determined by a one-way ANOVA, multiple comparisons. Sample size: n = 4.

**Figure 2 microorganisms-11-02131-f002:**
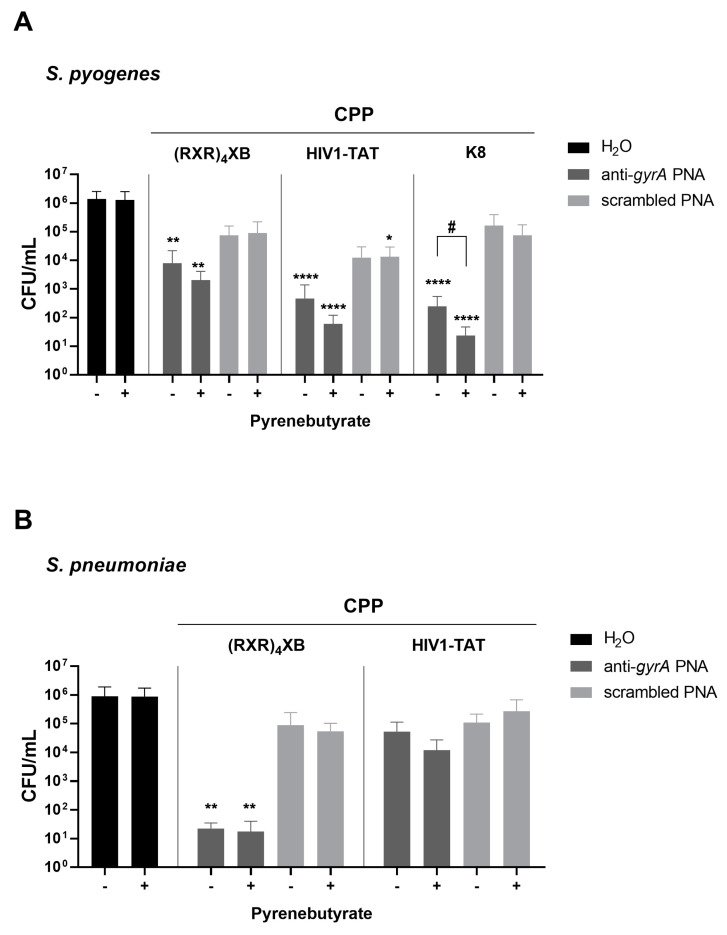
Reduction in bacterial CFU/mL following treatment with CPP-antisense PNAs for 6 h with or without pre-treatment with PyB. (**A**) Bacterial counts following treatment of *S. pyogenes* with 5 µM CPP-anti-*gyrA* PNA or 5 µM CPP-anti-*gyrA* scPNAs. (**B**) Bacterial counts following treatment of *S. pneumoniae* with 10 µM CPP-anti-*gyrA* PNA or 10 µM CPP-anti-*gyrA* scPNAs. Data are presented as mean values with standard deviation. Statistical significance was determined by a one-way ANOVA, multiple comparisons. Differences between PNA conjugate samples and mock control (untreated) were expressed as *p* ≤ 0.05 (*), *p* ≤ 0.01 (**) or *p* ≤ 0.0001 (****). Differences between PNA conjugate samples in the presence of PyB and PNA conjugate samples without PyB were expressed as *p* ≤ 0.05 (#). Sample size: *n* = 5.

**Figure 3 microorganisms-11-02131-f003:**
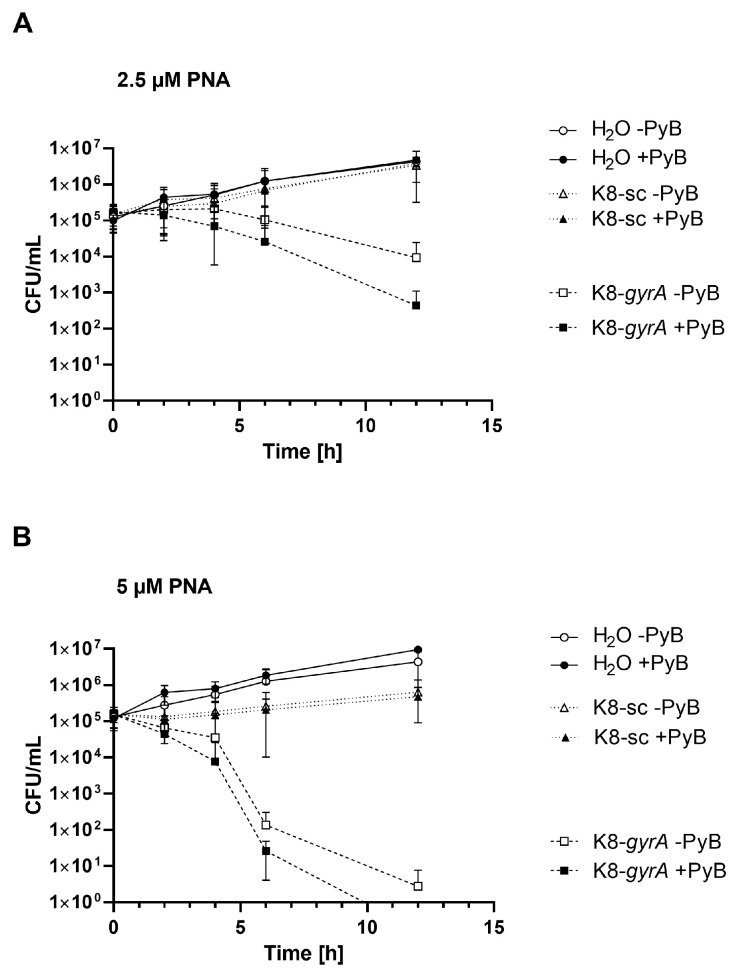
Killing kinetics of CPP-anti-*gyrA* PNA treatment in *S. pyogenes* with or without pre-treatment with PyB. (**A**) Bacterial counts following treatment with 2.5 μM K8-anti-*gyrA* PNA or 2.5 μM K8-anti-*gyrA* scPNA, respectively. (**B**) Bacterial counts following treatment with 5 μM K8-anti-*gyrA* PNA or 5 μM K8-anti-*gyrA* scPNA, respectively. Data are presented as means with standard deviation. Sample size: *n* = 3.

**Figure 4 microorganisms-11-02131-f004:**
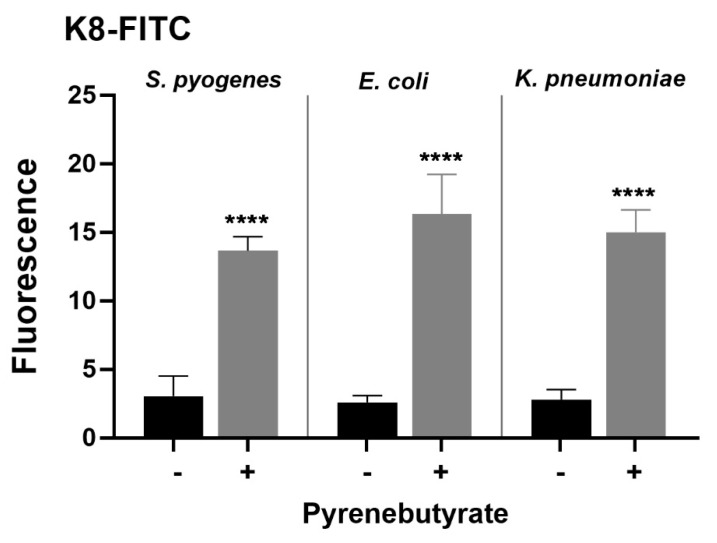
Transport of K8-FITC peptides into bacteria with or without pre-treatment with PyB. Fluorescence intensity of bacterial cell lysates following incubation with 10 μM K8-FITC. Data are presented as means with standard deviation. *p* ≤ 0.0001 (****). Sample size: *n* = 3.

**Table 1 microorganisms-11-02131-t001:** CPP–PNA conjugates for antisense-studies.

CPP	CPP–PNA Designation	CPP–PNA Sequence
*Streptococcus pyogenes*		
(RXR)_4_XB	(RXR)_4_XB-anti-*gyrA* PNA	RXRRXRRXRRXRXB-eg ^1^ -tgcatttaag
	(RXR)_4_XB-anti-*gyrA* scPNA	RXRRXRRXRRXRXB-eg-attagactgt
HIV-1 TAT (48–57)	TAT-anti-*gyrA* PNA	GRKKRRQRRRYK-eg-tgcatttaag
	TAT-anti-*gyrA* scPNA	GRKKRRQRRRYK-eg-attagactgt
Oligolysin (K8)	K8-anti-*gyrA* PNA	KKKKKKKK-eg-tgcatttaag
	K8-anti-*gyrA* scPNA	KKKKKKKK-eg-attagactgt
*Streptococcus pneumoniae*		
(RXR)_4_XB	(RXR)_4_XB-anti-*gyrA* PNA	RXRRXRRXRRXRXB-eg-tgcattaata
	(RXR)_4_XB-anti-*gyrA* scPNA	RXRRXRRXRRXRXB-eg-aatgattact
HIV-1 TAT (48–57)	TAT-anti-*gyrA* PNA	GRKKRRQRRRYK-eg-tgcattaata
	TAT-anti-*gyrA* scPNA	GRKKRRQRRRYK-eg-aatgattact
PenArg	PenArg-anti-*gyrA* PNA	RQIRIWFQNRRMRWRR-eg-tgcattaata
R9	R9-anti-*gyrA* PNA	RRRRRRRRR-eg-tgcattaata

^1^ eg: 8-amino-3,6-Dioxaoctan acid; X: 6-Aminohexanoic acid; B: β-Alanin.

**Table 2 microorganisms-11-02131-t002:** Positive amino acid residues per CPP.

CPP	Arginine Residues	Lysine Residues	Proportion of Positively Charged Amino Acids
PenArg	4	5	7/16 (44%)
(RXR)_4_XB	8	0	8/14 (57%)
TAT	6	3	9/12 (75%)
R9	9	0	9/9 (100%)
K8	0	8	8/8 (100%)

## Data Availability

The data presented in this study are available within the article.

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
