# Peer review of "Pyrenebutyrate Enhances the Antibacterial Effect of Peptide-Coupled Antisense Peptide Nucleic Acids in Streptococcus pyogenes"

_microorganisms, 2023, doi:10.3390/microorganisms11092131_

Round 1

Reviewer 1 Report

Lines 14, 38 and 78, review and remove spaces.

Line 28, correct to tN‐(2‐aminoethyl)glycine

Some clarifications should also be included, especially the use of PyB. In the introduction, line 58, the authors indicate "...4-(1-pyrenyl)-butyric acid, also known as pyrenebutyrate (PyB)...", but chemically the acid is different from the carboxylate ion, its conjugate base.

In the methodology, it is not clear how the 8-amino-3,6-dioxaoctan acid is acquired?, How are covalent bonds made? for PNA synthesis, and and how intramolecular reaction is avoided.

Better detail the preparation of PyB solution.

Lines 136 and 139, italicize the names of the bacteria.

Line 169, translate 6-Aminohexansäure.

In Table 2, what does the sequence of numbers from 1 to 24 mean?

Author Response

Reply to the Review Report Reviewer 1

Thank you for your helpful comments. We addressed the issues as follows:

>Lines 14, 38 and 78, review and remove spaces.

We removed the respective spaces

>Line 28, correct to tN‐(2‐aminoethyl)glycine

We corrected the term to N-(2-aminoethyl)glycine

> line 58, the authors indicate "...4-(1-pyrenyl)-butyric acid, also known as pyrenebutyrate (PyB)...", but chemically the acid is different from the carboxylate ion, its conjugate base

4-(1-pyrenyl)-butyric acid was dissolved and was active in solution as its ion. We rephrased the sentence to: One example that has already been studied for this approach is pyrenebutyrate (PyB) that is obtained from 4-(1-pyrenyl)-butyric acid.

> In the methodology, it is not clear how the 8-amino-3,6-dioxaoctan acid is acquired?, How are covalent bonds made? for PNA synthesis, and and how intramolecular reaction is avoided.

 We added the information to the text: CPP-PNAs were synthesized on a RinkAM resin using Fmoc chemistry followed by HPLC purification (Peps4LS GmbH, Heidelberg, Germany) [20]. The ethylene glycol linker (Fmoc-8-amino-3,6-dioxaoctanoic acid) and the respective Fmoc-protected amino acids for the CPPs were coupled directly to the PNA part on the resin.

>Better detail the preparation of PyB solution.

The description of the PyB solution preparation has been added to the text: 4-(1-pyrenyl)-butyric acid (Merck, Darmstadt, DE) was dissolved to 1 M in DMSO. The solution was stored in the dark and further dilution steps were performed in PBS, 3 % DMSO. 50 µl PyB in PBS/DMSO were added to the bacterial suspension at a final concentration of 50-400 µM and pre-incubated for 10 min at 37°C and 7 rpm.

> Lines 136 and 139, italicize the names of the bacteria.

We italicized the names and apologize for the sloppiness.

>Line 169, translate 6-Aminohexansäure.

We translated 6-Aminohexansäure to 6-Aminohexanoic acid

>In Table 2, what does the sequence of numbers from 1 to 24 mean?

I do not know. These numbers were not included in the table that was initially submitted. I assume that a formatting problem occurred following submission. I replaced the table in the manuscript with our original table.

Reviewer 2 Report

The manuscript by Patenge describes the effects that pyrenebutyrate has on antibacterial data observed with peptide-coupled antisense peptide nucleic acid in bacteria. The work is well presented, the data is solid, the methods are clear and the values support and validate their study. This work merits publication after the following minor issues.

1. Fix the units on all figures, it should be mL, no ml.

2.  Add a heading for the conclusion. 

3. Add references to the following sentence in the conclusion section: Line 341_ "Together with data from previous studies in eukaryotic cells and artificial membranes,.."

The quality of the English language is fine.

Author Response

Thank you for your helpful comments. We changed the manuscript accordingly:

  1. Fix the units on all figures, it should be mL, no ml.

We changed the units on all figures to “mL”

  1. Add a heading for the conclusion.

The heading has been added

  1. Add references to the following sentence in the conclusion section: Line 341_ "Together with data from previous studies in eukaryotic cells and artificial membranes,.."

We added these three references, that we had discussed before on this topic: Together with data from previous studies in eukaryotic cells and artificial membranes [14, 16, 17], these results indicate that PyB directly interacts with the bacterial membrane and thereby facilitates CPP-uptake.

  1. Perret, F.; Nishihara, M.; Takeuchi, T.; Futaki, S.; Lazar, A.N.; Coleman, A.W.; Sakai, N.; Matile, S. Anionic fullerenes, calixarenes, coronenes, and pyrenes as activators of oligo/polyarginines in model membranes and live cells. Journal of the American Chemical Society. 2005, 127:1114-1115.
  2. Guterstam, P.; Madani, F.; Hirose, H.; Takeuchi, T.; Futaki, S.; El Andaloussi, S.; Graslund, A.; Langel, U. Elucidating cell-penetrating peptide mechanisms of action for membrane interaction, cellular uptake, and translocation utilizing the hydrophobic counter-anion pyrenebutyrate. Biochim Biophys Acta. 2009, 1788:2509-17.
  3. Takeuchi, T.; Kosuge, M.; Tadokoro, A.; Sugiura, Y.; Nishi, M.; Kawata, M.; Sakai, N.; Matile, S.; Futaki, S. Direct and rapid cytosolic delivery using cell-penetrating peptides mediated by pyrenebutyrate. Acs Chemical Biology. 2006, 1:299-303.